# The Effects of Physical Activity on the Gut Microbiota and the Gut–Brain Axis in Preclinical and Human Models: A Narrative Review

**DOI:** 10.3390/nu14163293

**Published:** 2022-08-11

**Authors:** Stefania Cataldi, Luca Poli, Fatma Neşe Şahin, Antonino Patti, Luigi Santacroce, Antonino Bianco, Gianpiero Greco, Barbara Ghinassi, Angela Di Baldassarre, Francesco Fischetti

**Affiliations:** 1Department of Basic Medical Sciences, Neuroscience and Sense Organs, University of Study of Bari, 70124 Bari, Italy; 2Department of Sport and Health, Faculty of Sport Sciences, Ankara University, Ankara 06830, Turkey; 3Sport and Exercise Sciences Research Unit, Department of Psychology, Educational Science and Human Movement, University of Palermo, 90144 Palermo, Italy; 4Interdisciplinary Department of Medicine—Section of Microbiology and Virology, Policlinico University Hospital, University of Study of Bari, 70124 Bari, Italy; 5Department of Medicine and Aging Sciences, G. D’Annunzio University of Chieti and Pescara, 66100 Chieti, Italy

**Keywords:** cognitive functions, anxiety, depression, microbiome, overweight, elderly, athletes, sports, overtraining, fitness

## Abstract

Increasing evidence supports the importance of the gut microbiota (GM) in regulating multiple functions related to host physical health and, more recently, through the gut–brain axis (GBA), mental health. Similarly, the literature on the impact of physical activity (PA), including exercise, on GM and GBA is growing. Therefore, this narrative review summarizes and critically appraises the existing literature that delves into the benefits or adverse effects produced by PA on physical and mental health status through modifications of the GM, highlighting differences and similarities between preclinical and human studies. The same exercise in animal models, whether performed voluntarily or forced, has different effects on the GM, just as, in humans, intense endurance exercise can have a negative influence. In humans and animals, only aerobic PA seems able to modify the composition of the GM, whereas cardiovascular fitness appears related to specific microbial taxa or metabolites that promote a state of physical health. The PA favors bacterial strains that can promote physical performance and that can induce beneficial changes in the brain. Currently, it seems useful to prioritize aerobic activities at a moderate and not prolonged intensity. There may be greater benefits if PA is undertaken from a young age and the effects on the GM seem to gradually disappear when the activity is stopped. The PA produces modifications in the GM that can mediate and induce mental health benefits.

## 1. Introduction

In recent years, research on the interaction between gut microbiota (GM) and health has expanded considerably. High-throughput sequencing has favored and facilitated research on GM in multiple aspects [1]. Indeed, humans have specific microbial profiles at the oral, skin, reproductive organ, gastrointestinal and fecal levels [2,3]. In particular, the human gastrointestinal (GI) tract has a microbial diversity of up to 100 trillion microorganisms, predominantly bacteria but also archaea, viruses, and parasites. The GM encodes more than three million genes that produce hundreds of metabolites [4]. This vast ecosystem is susceptible to adaptive changes based on multiple factors, external and internal, such as genetics, age, diet, drugs, stress, physical activity (PA), including exercise [5,6]. Only recently, the effect of the PA on the GM has been more thoroughly investigated, although considerable difficulties persist in structuring research protocols capable of limiting the various confounding elements as much as possible.

Most evidence shows that a sedentary lifestyle is associated with a higher incidence of chronic diseases [7,8,9], on which PA can play a preventive and therapeutic role through different mechanisms. Recently, the potential effects produced on GM have been proposed as another modality through which PA can perform beneficial functions on host health [10]. In fact, the proliferation of some bacterial strains is also possible through PA, but it seems that a basic GM remains stable from adulthood to elderhood [3,4], while it would appear that between 20–60% of bacterial diversity could be influenced by diet and PA [11,12,13,14].

Although there is no gold standard for gut health, and its definition remains uncertain [15,16,17], the interaction between host PA can favor the growth of beneficial bacteria and gut cells that maintain the integrity of the intestinal surface, which acts as the first defense against pathogens. In return, the selected bacteria synthesize molecules capable of modifying the metabolism and immune functions of the host [18]. Furthermore, disruption of the gut barrier is considered one of the mechanisms of major depression and other cognitive and behavioral alterations, with the gut–brain axis (GBA) playing a role in mediating these processes [19]. In this context, PA seems able to influence this two-way communication interaction through the GBA [20].

This review aims to summarize the current evidence about the effects that PA has on GM and how GM, in turn, can influence physical performance and cognitive functionality, providing an overview of the currently most promising aspects in this field of research. We first discuss the extent to which GM can affect the physiological condition of its host and its influence on the onset of certain pathologies when an imbalance state occurs. Next, we analyze the evidence from preclinical studies concerning different PA modalities, the effects of PA on young or elderly subjects, as well as the interaction between PA and GM-mediated physical performance and overtraining. We then review the influence of PA on the human intestinal microbiota, specifically the role of physical fitness, the impact of endurance activity, and the effect that PA can exert on the microbiota of the elderly population; lastly, we discuss the relationship between PA and GM in the overweight subjects. Finally, we look at how and to what degree PA may exert certain effects on the GBA and indirectly on cognitive function.

## 2. Materials and Methods

This narrative review was carried out following the Narrative Review checklist [21]. Comprehensive research of Medline In-Process and other Non-Indexed Citations was conducted (prior to 31 March 2022), using MEDLINE(PubMed), Web of Science, and Google Scholar to retrieve relevant articles. The literature search was made using medical subject headings (MeSH) and Boolean syntax. Controlled terms were used to search for studies (“Physical Activity” OR “Physical Exercise” AND “Microbiota” OR “Microbiome”)/(“Sport” OR “Athletes” AND “Microbiota” OR “Microbiome”)/(“Gut-brain” OR “GBA” AND “Physical activity” OR “Physical exercise” AND “Microbiota” OR “Microbiome”). Following filters were used: Full text, pre-clinic or human studies, and English language. After candidate articles were collected, further identification was conducted based on inclusion and exclusion criteria.

### Studies Selection

The inclusion criteria were only English-language original peer-reviewed articles, randomized and non-randomized studies, and observational and pilot studies published from January 2008 to March 2022. Excluded records were review articles, meta-analyses, practical guidelines, unpublished studies, editorials, letters to the editor, and essays, although they were used as an added measure to ensure search comprehensiveness and used as references.

We included pre-clinic (murine models) studies except for those involving nutritional supplementation to enhance exercise performance or pre/pro-biotics and/or antibiotics (in the 2 months before the intervention), and human studies with the exception of those that included subjects who take or have taken (in the 2-months before the intervention) pre/pro-biotics and/or antibiotics, as well as those that provided a specific sports supplementation intervention since evidence suggests that these conditions may lead to significant changes in the composition of gut microbiota. The search was not restricted to the frequency, intensity, type, or time (FITT) of exercise, gender, age, clinical condition, sample size, or length of follow-up.

Since human and animal studies analyzing the involvement of the gut microbiota in various responses to PA and exercise are still a young field of research, especially about the interaction between these and the GBA, we did not use restrictive criteria for selecting the papers for this narrative review.

Two team authors independently extracted relevant information from included studies: author, year of publication, study design, number and age of participants, species, type of PA carried out, protocol and diet assessment, duration of intervention, and main outcomes. Any disagreements were resolved by consensus. The characteristics of the studies were summarized, and data on the effects of PA on the metabolic variables of GM were synthesized.

## 3. Results

### 3.1. Identification of Studies

At the end of the selection process, 1907 articles were extracted, of which *n* = 823 from Web of Science, *n* = 727 from PubMed, and *n* = 357 from Google Scholar. Each title and abstract were screened for relevance, removing review articles, unpublished studies, meta-analyses, practical guidelines, editorials, letters to the editor, and essays (*n* = 876). Thereafter, the search strategy was based on the assessment of the full text of the remaining 121 articles to verify their eligibility. Lastly, on a total of 46 research articles selected, 19 focused on pre-clinic (murine) models and 27 on humans, specifically focusing on GM responses to PA, were included (Figure 1).

### 3.2. Studies Characteristics

Thus, this narrative review provides an evaluation of 46 studies [22,23,24,25,26,27,28,29,30,31,32,33,34,35,36,37,38,39,40,41,42,43,44,45,46,47,48,49,50,51,52,53,54,55,56,57,58,59,60,61,62,63,64,65,66,67] inquiring in-depth about the PA-induced changes on GM, and the resulting physical and cognitive effects, in pre-clinic [49,50,51,52,53,54,55,56,57,58,59,60,61,62,63,64,65,66,67] and human models [22,23,24,25,26,27,28,29,30,31,32,33,34,35,36,37,38,39,40,41,42,43,44,45,46,47,48]. Most of them (*n* = 26) [23,24,26,28,31,34,38,39,41,49,50,51,52,53,54,55,57,58,59,60,61,62,64,65,66,67] regarded GM modifications following aerobic PA, while other works (*n* = 9) [29,32,33,35,36,46,48,56,63] focused on changes occurring after an aerobic endurance protocol, few studies (*n* = 3) [40,44,47] focus on the changes induced by concurrent training or (*n* = 3) [37,43,45] separate interventions of resistance and aerobic exercises, lastly, further works (*n* = 5) [22,25,27,30,42] investigated the effects produced on the GM following the practice of specific sports. Table 1 and Table 2 summarize the characteristics of the human and pre-clinic studies included.

## 4. Discussion

### 4.1. The Gut Microbiota

Over millennia, human beings evolved symbiotically in environments where there was enormous microbic diversity [68]. The endosymbiosis theory even suggests that today’s eukaryotic cells were formed through the union of simpler cells, symbiotic bacteria, capable of performing certain specific functions on their own, such as extracting energy from organic compounds through aerobic respiration mechanisms. These conditions meant that over time, the bacteria themselves became completely dependent on the cell that had incorporated them, and similarly, the latter obtained an extra supply of energy in return [69,70,71].

Only in recent decades our knowledge of microbes cohabiting with humans has advanced to such an extent that we are able to make more concrete hypotheses about their functions and interactions with the host, as well as the mechanisms through which they are able to produce their effects. Since 2001, when Joshua Lederberg [72] came up with the term microbiome, studies on this subject have proliferated thanks also to new technologies capable of investigating this area. Especially since 2006, when Gill et al. [73] succeeded in sequencing the genome of human bacteria for the first time, demonstrating the presence of an interaction with lifestyle, diet, age, and other factors with the human intestinal bacterial composition. We now know that our bodies, and in particular our digestive tracts, are populated by a huge number of bacteria (microbiota) with which we have a symbiotic relationship that has a profound effect on our quality of life and whose gene expression (microbiome) is a second set of genes. The evolution of scientific knowledge has led us to look at our bodies no longer as simple organisms but rather as super-organisms that functions thanks to the perfect balance of a mixed genetic component, composed partly of human genes and partly of bacterial genes. Every time a part of these bacteria is destroyed, the relative balances within this ecosystem are altered and remain altered, sometimes for months or years. This results in a series of serious physiological imbalances that, in the long run, can give rise to the so-called diseases of progress [74].

The GM has important roles in the training of host immunity, digesting food, and endocrine and neurological functions, producing numerous compounds that influence the host. Thus, it appears that a considerable part of the environmental influence on human health and disease risk may be mediated or modified by microbial communities [75]. Alterations in the gut environment can affect GM community composition and function [76,77]. Through these pathways, the GM communicates with the host, affects host systems and organs, and modulates the host’s physiological functions, including glucose metabolism and liver function. Alteration in the GM composition can potentially be associated with various chronic diseases, such as allergic asthma, inflammatory bowel disease (IBD), major depression, and obesity [78].

New sequence technologies, together with measures of host physiology and mechanistic experiments in humans, animals, and cells, hold potential as initial steps in the identification of potential molecular mechanisms behind reported associations [79].

### 4.2. Physical Activity and Gut Microbiota in Preclinic Studies

It is now quite clear that, in animal models, exercise initiates significant changes in the GM. Matsumoto et al. [49], in a first study, investigated the effect of exercise on the gut microbiota, showing that after five weeks, mice given free access to the exercise wheel (VWR) had an altered microbiota composition compared with the control group of sedentary mice (SED); this was accompanied by the detection of changes in two species of butyrate-producing bacteria (*SM7/11* and *T2-87*) with a relative twofold increase in butyrate concentration induced by exercise. This is important because butyrate, a short-chain fatty acid (SCFA), is the preferred energy source for epithelial cells lining the colon and has been shown to have multiple beneficial functions on gut function, including regulation of satiety, insulin sensitivity, and inflammation [80].

Allen et al. [55] found that voluntary exercise attenuates and forced exercise increases intestinal inflammation in a model of colitis, hypothesizing that these different exercise patterns may differentially influence GM. Choi et al. [50] evaluated the effects of six weeks of VWR on the microbiota by comparing healthy mice and mice given polychlorinated biphenyls (PCBs), carcinogenic toxins with adverse modulatory effects on gut function [81,82], some of which are currently banned due to their dioxin-like toxicological properties and risk of bioaccumulation in humans [83]. They found that exercise significantly attenuated the changes produced in the GM by PCBs, as demonstrated by a greater bacterial diversity in mice given PCBs after voluntary exercise for five weeks than in the group of sedentary mice that still received PCBs. Specifically, 67 taxa, groups of bacteria with similar morphological, biochemical, and genetic aspects, were found only in mice that exercised. Of these, half are from the *Lactobacillales* order, which demonstrates beneficial properties for colon health and immune function [84,85] as well as improved protective function in intestinal disorders [86,87]. These results could be related to experiments in germ-free (GF) mice showing that GM can regulate the expression of cytochrome *P450*, an enzyme involved in the metabolism of various foreign substances, including chemical pollutants [88].

Kang et al. [52] demonstrated that forced exercise (FWR), required to precisely control training volume, alters several groups of bacteria in both mice fed a normal diet and those on a high-fat diet (HFD), the latter of which is known to induce changes in the bacterial community and metabolic state of the host [89]. Similar results were found by Evans et al. [53], who sought to investigate whether exercise (VWR) can alter the GM in HFD-fed subjects. Specifically, this work observed the modification of multiple classes and families of bacteria following exercise, even in the group given an HFD. At a more general taxonomic level, exercise has shown the ability to increase the ratio of *Bacteroidetes:Firmicutes*, the main phyla in humans [90], where *Bacteroidetes* are directly associated with lean mass, and therefore, in the case of obesity, there is a reduction, and in parallel when obese subjects lose weight there is an increase [91]. In addition, according to Matsumoto et al. [49], Evans et al. [53] found a significant growth of butyrate-producing bacteria (e.g., *Bacteroidales S24-7*, *Clostridiaceae*, *Lachnospiraceae*, and *Ruminococcaceae*). Similar conclusions were reached by Campbell et al. [59] using a 12-week protocol and randomly assigning experimental subjects to four groups: lean sedentary (LS), sedentary (OS) with diet-induced obesity (DIO), lean active (LX) and DIO active (OX), with controlled caloric intake. They were able to observe that exercise (FWR) appears to influence the microbiota independently of diet. In this case, however, GM changes were analyzed at the genus level rather than at the phylum, class, or family level, highlighting the presence in active mice of bacteria related to *Fecalibacterium prausnitzii*, which may provide intestinal protection.

In agreement with Kang et al. [52], Denou et al. [60] also found an increased bacterial diversity in DIO mice fed an HFD following forced treadmill running (FTR); a proliferation of *Bacteroidetes* was observed in the cecum, colon, and feces with a consequent reduction in the *Firmicutes:Bacteroidetes* ratio. In contrast to Kang et al. [52], using a high-intensity interval training (HIIT) protocol did not result in a reduction in body mass, suggesting the possibility of different bacterial changes based on the exercise modality used. The data from this study suggest that exercise can counterbalance the changes made to the GM by obesity, but how this may occur is not entirely clear. Increased intestinal motility and reduced colonic blood flow during exercise are some of the possible modalities investigated.

Ribeiro et al. [64] highlighted that the use of a medium-low intensity exercise (50% of maximal velocity) induces insignificant changes in the GM to counteract the effects of a prolonged HFD.

### 4.3. Voluntary or Forced Exercise?

It has been found [55] that FWR results in a significant up-regulation of the *Ruminococcus gnavus* species, a bacterium capable of degrading the intestinal mucosa and thus penetrating its inner and outer layers, exposing intestinal epithelial cells to immunogenic bacterial proteins and thus exacerbating intestinal inflammation [92]. However, the regulation of mucin, the glycoprotein that makes up intestinal muscle, and the bacterial dynamics involved in relation to exercise have been little investigated. In the case of VWR, distinctive changes are induced both in the entire bacterial community and in individual taxa present in the GM of mice. Of note is the increase in the *Anaerotruncus* genus, a butyrate-producing bacterial group that colonizes the outer layer of the colonic mucosa and is phylogenetically related to *Fecalibacterium prausnitzii*, a known butyrate producer in humans [93,94]. These bacteria often feed on lactate, acetate, or other intermediates produced by other bacterial strains, and so it is possible that the activity-induced changes in so-called lactate producers (e.g., *lactobacilli*) and butyrate producers (e.g., *Anaerotruncus*) are related by a cross-feeding phenomenon.

Kellermayer et al. [95] also found a complete abolition of the *Turicibacter* population in so-called knockout mice, genetically modified not to express a given gene, for Toll-like receptor-2 (TLR-2), transmembrane molecules expressed in a number of intestinal cells (epithelial, B and T lymphocytes, macrophages) involved in the complex network of microbial pattern recognition associated with chronic intestinal inflammation disorders, although its role as a regulator of intestinal inflammation is still debated. Both papers thus suggest an immune and disease regulatory capacity of *Turicibacter*. This is related to exercise by the previously cited work of Evans et al. [53], which shows that VWR significantly reduces *Turicibacter* species populations in feces and caecum, indicating a possible immune-regulatory role of exercise through changes in the GM.

### 4.4. Exercise and Host Age

The period of life in which exercise is carried out would also appear to be important, thus representing age as a determining factor in the regulation of the GM. Indeed, some research has shown that the impact of GM on host physiology can be age-dependent; using GF mice, an early sensitive period has been identified during which the absence of an intact GM reflects physiological consequences such as the exaggerated activity of the hypothalamic-pituitary-adrenal (HPA) axis that can only be partly normalized by the introduction of *Bifidobacteria infantis* if administered during the early period of life [96].

Just as early exposure to certain microorganisms (e.g., *Bifidobacteria* species) can program the future immune system, it has recently been shown, in mice, that administration of antibiotics in the early period of life leads to an increase in total body mass as well as fat mass in adulthood [97]. This evidence reveals that the framework of GM during development can strongly influence the health of the host throughout life. Furthermore, just as physiological systems during development are remarkably malleable and sensitive to change, likewise the GM is more plastic in the early stages of life [98,99] and, consequently, the microbial ecosystem present during the early period of life may be more sensitive to environmental changes due in part to its lower stability and diversity than in adulthood. Thus, manipulations of environmental factors could have a more important impact on the GM framework if implemented early in life. Obviously, PA ranks well among those modifiable environmental factors, and so Mika et al. [58] hypothesize and note that exercise undertaken during early life (EL) may have a greater impact on GM than that undertaken in adulthood, corroborated recently by another study [100]. This is highlighted by comparison at the phylum level, where changes in phyla are only observed in young rats, and at the genus level, where a more important change in the microbial genus is observed in young than in adults. In terms of phylum, exercise in EL subjects increases the presence of *Bacteroidetes*, associated with a lean phenotype [101], while decreasing that of *Firmicutes*, which are correlated to obesity [52]. In this work, therefore, microbial patterns reflecting a lean phenotype were only observed in EL subjects indicating that the developmental stage during which exercise is performed may be important in determining this adaptive change in phyla. The absence of changes under the phylum profile in adult subjects, however, appears to contrast with the observation by Clarke et al. [22] in humans, Petriz et al. [54], and Evans et al. [53] in rats. Regarding the genus, six modified genera are identified in EL subjects and three in adults. In young mice, there is an increase in species *Bifidobacteria* and *Methanosphaera*, the latter belonging to the Archaea domain, which, by using hydrogen as an energy source [102], may help to make carbohydrate fermentation more efficient and less problematic. This study, therefore, suggests that although six weeks of exercise initiated in adulthood may alter the abundance of some bacterial genera, if the same activity is carried out in early life, there is a greater capacity to influence the overall structure of the microbial ecosystem. Furthermore, these changes in phylum and genus appear to be consistent with phenotypic changes in the body composition of EL subjects, who report an increase in lean mass, unlike adult rats, which agrees with the observations of Shindo et al. [103]. Nevertheless, it is emphasized that this effect might simply be the direct result of exercise, although the reported data set suggests that GM composition in young rats might play a role in promoting and maintaining gains in lean mass.

Measurement of α-*diversity* measures the variation of microbes in a single sample, i.e., the number of species (taxa) present (richness) and whether certain species are homogeneous or dominant (diversity) in the sample. Through this assessment, Mika et al. [58] observe a lower species richness in juvenile than in adult rats. A similar condition was observed in humans by Yatsunenko et al. [104], supporting the idea that it is precisely this condition of the young GM that encourages greater changes within it, unlike in the adult GM where the greater microbial complexity may make it more resistant to changes induced by environmental factors. In addition, they observe a lower Shannon value, which indicates a lower homogeneity in the microbial structure; the Shannon evenness index measures how homogeneously the bacteria are distributed in a specific sample without considering the number of species, so the lower this value is, the less homogeneous the microbial population will be. This, together with lower species richness, indicates that exercise in EL subjects can affect the uniformity of the bacterial community by selecting specific taxa. However, it should be noted that these results contrast with the findings of Clarke et al. [22] in humans and Petriz et al. [54] in several rat species. Furthermore, although this is the first work that attempts to investigate a relationship between exercise, GM, and age, the measurement of bacterial composition only in stool samples, the failure to control for the caloric intake of subjects, and the use of separate cohorts represent caveats that do not allow a more concrete causality to be extrapolated from the simple correlation.

### 4.5. Exercise, Performance and Overtraining Mediated by Gut Microbiota

Hsu et al. [56] sought to establish a relationship between performance and physical fatigue, antioxidant enzyme levels, and GM by comparing three groups: GF mice, GF mice colonized by a single genus of bacterium *Bacteroides fragilis* (BF), which is also normally present in the GM of the human colon, and pathogen-specific deprived (SPF) mice with an intact microbiota. By administering the endurance swimming test, commonly used to assess the degree of fatigue in animal models, the best time was obtained by SPF mice, with complete microbiota, followed by BF and finally GF mice. At the same time, a significant reduction in the activity of antioxidant enzymes was observed in GF mice, suggesting the influence of GM on the latter, which in turn can influence physical performance, as similarly reported by other studies [105,106]. A lower activity of glutathione peroxidase (GPx) and catalase (CAT), antioxidant enzymes that may contribute to counteracting the onset of fatigue, was found in GF mice compared to SPF and BF mice; therefore, the absence or limited presence of a bacterial community could be associated with an under-regulation of GPx and CAT activity, thus affecting physical performance.

Queipo-Ortuño et al. [51] evaluated differences in GM composition in rats during different nutritional conditions and PA and identified their relationship with leptin and ghrelin levels. The protocol involved dividing subjects into four groups: calorie restriction combined with voluntary PA (ABA), calorie restriction in the absence of PA (control ABA), ad libitum feeding with exercise (EG), and ad libitum feeding without PA (AL). Thus, there was a significant increase in *Proteobacteria*, *Bacteroides*, *Clostridium*, *Enterococcus*, *Prevotella,* and *M.smithii* and a consequent reduction in *Actinobacteria*, *Firmicutes*, and *B. coccoides—E. rectale*, *Lactobacillus,* and *Bifidobacterium* in the groups subjected to calorie restriction compared to those on ad libitum feeding. In particular, the increase in *Prevotella* species could be related to the degradation and reduction of mucin, which is essential for the integrity of the intestinal mucosa. On the other hand, in the EG group, *Lactobacillus* and *Bifidobacterium* genera increased, both capable of producing lactate, which is then converted into butyrate by specific GM bacteria. Thus, the potential negative impact on the number of bacteria beneficial to the health of the host and the increase in those potentially related to the destruction of the gastrointestinal barrier in subjects on calorie restriction accompanied by PA is highlighted. Although the researchers used male rats of the same weight from the same farm to reduce GM variation, the different types of diet during the protocol remain an important confounder when trying to extrapolate conclusions about the impact of PA on GM. Concordant, Petriz et al. [54], using a training protocol with a slightly shorter duration, four weeks, detected an increase in the *Lactobacillus* genus in obese rats following exercise. Recalling that this type of bacteria (LAB) in the gastrointestinal tract would appear to induce beneficial effects by influencing the intestinal microbiota, protecting the mucosa, and competing with pathogens [107], it is suggested that exercise can modify the GM at the level of the bacterial genus, with possible consequent beneficial effects, particularly in obese subjects.

Liu et al. [57], instead, sought to understand whether PA (VWR) could have an impact on the GM in ovariectomized (OVX) female rats fed with HFD. Specifically, as aerobic capacity is largely genetically predetermined [108], rats genetically selected to express high (HCR) or low (LCR) running capacity were used to assess how intrinsic fitness status might influence the impact of exercise on GM. Subjects were ovariectomized to reproduce a pattern like that established in post-menopausal women. The higher caloric intake and greater distance traveled found in the HCR group are associated with a relative abundance of *Christensenellaceae* and a reduced abundance of *Clostridiaceae* and *Desulfovibrionaceae*. In contrast to Turnbaugh et al. [89], HCR subjects show reduced α-*diversity* compared to the LCR group, while similar to Petriz et al. [54], an increase in the relative abundance of *Firmicutes* and reduction of *Proteobacteria* in HCR rats after exercise is noted. In the LCR group, as already observed [53], the opposite situation occurs. However, the relative abundance of *Bacteroidetes* appears to be unchanged within the different groups, contrary to the work of Petriz et al. [54] and Evans et al. [53]. It was observed that eleven weeks of VWR changed the relative abundance of different taxa at the family and genus level and that the type of this change depended on the genetic line (HCR vs. LCR). Due to the multiple variables considered to which altered hormonal balance is added, it becomes difficult to extrapolate conclusions regarding the exclusive impact of exercise on GM.

Li et al. [67] note that HFDs, by reducing microbial diversity, enhancing the proliferation of a pro-inflammatory microbiota, and increasing intestinal permeability (IP), can induce an increase in circulating Lipopolysaccharide (LPS) levels, endotoxins present on the membrane of certain bacteria, which are related to the onset of osteoarthritis [109]. PA (VWR) seems able to remodel the GM by increasing its diversity and consequently reducing LPS levels in the blood and synovial fluid. According to this work, PA, in addition to reducing body weight, may have a protective effect on cartilage by modifying the GM and reducing serum LPS levels.

Leigh et al. [66], in agreement with Lamoureux et al. [61] and Ribeiro et al. [64], after four weeks of moderate treadmill exercise, found no relevant changes in the overall composition of the GM. Specifically, when attempting to investigate the relationship between exercise and GM composition in rats exposed to two types of diet, they observed slight changes in the microbiota of those fed a standard chow diet but no changes to counteract the changes induced by a ‘cafeteria diet’ (CAF), a 10% sucrose solution along with commercial products such as biscuits and salty foods. These results may also be due in part to the use of this type of diet (CAF), which results in greater metabolic changes than the classic purified HFD used in the other studies.

Yuan et al. [63] set out to assess the negative effects of excessive exercise on the immunity system, energy metabolism, and GM. After four weeks of the protocol (excessive ES swimming), they found reduced α-*diversity* and β-*diversity* compared with the control group. In terms of phylum, the ES group showed an abundance of *Bacteroidetes* and *Firmicutes* at the ileocecal level and a reduction in *Proteobacteria*. In the families, there is a lower presence of *Bacteroidales: S24-7* and *Lachnospiraceae* and an increase in *Helicobacteraceae*. At the genus level, there is an increase in *Helicobater* and *Bacteroides* and a reduction in *Odoribater*. It is thus observed that certain disease-related bacteria appear to be present to a greater extent in the ES group than in the control group. However, due to the limited sample, these differences do not have statistical significance.

### 4.6. Physical Activity and Gut Microbiota in Human Studies

Clarke et al. [22] performed the first study on athletes. Expecting to find greater bacterial diversity in athletes than in sedentary subjects, they set up the study by recruiting 40 male rugby players with an average BMI of 29.1 and, due to their size, two control groups were composed of sedentary subjects, one with a BMI ≤ 25 (LBG) and the other with a BMI > 28 (HBG). Stool and blood samples were collected from all participants. Thus, greater richness and α-*diversity* were observed in the athletes than in the control groups, in agreement with Mörkl et al. [25]. Microbial diversity is positively associated with protein intake, suggesting that exercise and diet are drivers of gut bacterial diversity. In the athletes’ group, an increase in butyrate-producer phylum (*Firmicutes*) and genus (*F. prausnitzii*) was observed [93,94]. Furthermore, athletes and LBG show increased *Akkermansiaceae* family and *Akkermansia* genus, associated with metabolic disorders [110]. Despite these interesting correlations between exercise and GM, the research appears confounded by its nature as a cross-sectional study and no control of confounders. Three years later, researchers involved in this work conducted a new study [30] through which they try to understand whether the taxonomic differences noted in their previous work [22] are reflected at the functional level. They re-examined the GM of professional rugby players during pre-season training, investigating the metagenome. They observed, in athletes, the strengthening of certain pathways, which express a beneficial potential for the host, such as those for amino acids (AA) biosynthesis and carbohydrate metabolism. Finally, there is a high concentration of SCFAs in the feces.

Bressa et al. [24] found no modification in GM richness and diversity (α and β) in people who perform 30 min of moderate-intensity PA three days a week [111]. A food frequency questionnaire (FFQ) is administered for nutritional assessment, and stool samples are collected for microbial assessment. No changes in the *Bacteroidetes:Firmicutes* ratio were found. A lower abundance of the family *Turicibacteraceae* and genus *Turicibacter* is observed in the active group, as already found in animal studies [53,55,112], showing the first correlation between certain bacterial taxa and the introduction of exercise in humans. A significant abundance of *Coprococcus* genus and certain species (*R. hominis*, *A. muciniphila* and *F. prausnitzii*) in the active group compared to the sedentary group was observed, species associated with positive effects on health [113,114,115]. Although macronutrient and caloric intake were similar between the groups, different intakes of fruit/vegetables and animal products that can alter the GM [116] may confound the results.

Castellanos et al. [39], using data collected through their previous observational study [117] from a cohort of 109 volunteers (18–40 years old, BMI 20–30) classified as active, at least 3 h of activity per week, or sedentary, no minimum PA as defined by the World Health Organization (WHO) [111], analyzed the GM to identify the relevant bacteria involved in the reorganization of the microbial network. They identified bacterial taxa considered to be ‘key’ microorganisms for the structure of the GM since the pathogenicity of certain bacteria is not always and only related to their abundance but also to other factors such as interactions with other microorganisms. *Roseburia fecis* species, considered a health marker, *Rikenellaceae* and *Erysipelotrichaceae* families, whose role is not yet clear, were found in active subjects, while unclassified species of *Sutterella* genus, associated with impairment of cognitive and immune system function, were found in the sedentary group, suggesting that the GM of active subjects have higher efficiency.

No changes were found after three weeks of HIIT protocol, indicating that short-term HIIT protocol does not impact the fecal bacterial community and that progress in the cardiorespiratory fitness (CRF) does not lead to modification of GM in the short term [41], similarly to Moitinho-Silva [45] after six weeks of exercise

### 4.7. Influence of Physical Fitness on the Gut Microbiota

Estaki et al. [23] analyzed the composition of the GM as a function of the CRF, the overall capacity of the cardiovascular and respiratory systems that contributes to the ability to perform prolonged and strenuous exercise. To isolate, as far as possible, the influence of physical fitness on GM diversity, stool samples of subjects with different fitness levels, assessed by means of the cycle test for VO_2peak_, and a comparable diet were analyzed. After recruiting 39 healthy subjects (18–35 years) and interviewing them about their dietary habits (24 h dietary recall), they found that in individuals with higher CRF levels, bacterial diversity (α and β) is higher, regardless of diet. No single bacterial taxon or group of taxa showed significant variation in relation to CRF levels, but the functionality of the GM appears to favor an increase in chemotaxis-related genes, which are useful for bacterial nutritional processes, and a reduction in LPS biosynthesis pathways, indicating that microbial diversity and aerobic fitness are related. A study conducted by Yang et al. [26] on 71 premenopausal women (19–49 years), using a cycle ergometer to measure VO_2max_ and analyzing stool samples, came to similar conclusions. Low aerobic fitness correlates positively with a higher abundance of *Eubacterium rectale–Clostridium coccoides* (EreC), present in obese subjects [118], *Enterobacteria*, and a decrease in *Bacteroides*. However, these results lose relevance due to the various limitations present, first and foremost the percentage of fat mass of subjects with low aerobic fitness. These, in fact, have a higher BMI than subjects with a higher CRF, and once the results are corrected for this factor, the observed differences disappear.

A year later, Durk et al. [34] looked for a new correlation between CRF and GM composition in young (22–32 years) and healthy subjects. CFR is assessed by the Symptom-Limited Maximal Grade Treadmill Exercise Test to measure VO_2max_, while the dietary aspect was asked to follow the usual diet for seven days by tracking intake through MyFitnessPal.com. They observed a correlation between VO_2max_ and the *Firmicutes:Bacteroidetes* ratio. Although limitations like the collection of information on different variables, such as caloric intake, through subjective reports from participants, the results of this study support other similar work in both animal models and humans [23,26,55], indicating the PA as a factor that can positively affect the human GM. Similarly, Paulsen et al. [28], in a longitudinal study of breast cancer survivors using submaximal treadmill testing, found that CRF was associated with increased β-*diversity*, as also found by Estaki et al. [23] where, however, β-*diversity* was not explained by VO_2max_ but by protein intake. The nature of a pilot study and thus the small sample size is the main limitation.

Bycura et al. [43] examined the changes in the GM following a separate double intervention for the duration of eight weeks characterized by cardiorespiratory exercise (CRE) or resistance training exercise (RTE). A total of 28 subjects (18–26 years old) were assigned to the CRE group and 28 subjects (18–33 years old) to the RTE group. In contrast to Quiroga et al. [40], aerobic and resistance exercises were analyzed separately, observing that only aerobic activity causes an initial significant change within the GM that subsequently decreases until it becomes irrelevant.

### 4.8. Endurance Activities and Gut Microbiota

A first pilot study conducted by Petersen et al. [27], on 22 professional cyclists and 11 amateurs at a competitive level, investigated whether a difference in GM composition between professionals and amateurs can be detected through an analysis of the metagenome (representative of the species present) and the metatranscriptome (representative of the functions expressed in a specific environment). There was a significant correlation between the presence of the genus *Prevotella*, with concomitant upregulation of BCAAs, and the time spent training (>11 h/week) in both professionals and amateurs. A decrease in *Bacteroides* and, among 30 cyclists, an increase in the genus *Akkermansia* was noted, as already observed [22]. It was also observed that the increased presence of *Prevotella* correlates with certain carbohydrate and AAs metabolic pathways, including the biosynthesis of BCAAs [119]. However, as no FFQs were administered for dietary analysis, the abundance of *Prevotella* may be due, at least in part, to the high calorie and carbohydrate intake typical of a cyclist’s diet, coupled with the high weekly hours spent on training. In professional athletes, the metatranscriptomic analysis showed a high percentage of *M.smithii* compared to their amateur counterparts, associated with energy efficiency [120,121].

A second work by Scheiman et al. [35] was conducted on 15 marathon runners during the Boston Marathon and 10 sedentary subjects. A relative abundance of the species *Veillonella atypica* was observed, which can positively influence running performance through the conversion of lactate, produced by muscle activity, into propionate (SCFA), which is used for energy purposes. It has been shown that propionate can increase heart rate, VO_2max_, and influence blood pressure in mice [122,123] as well as increase resting energy expenditure and lipid oxidation in fasted humans [124]. While it was shown for the first time that GM could significantly contribute to physical performance and its benefits, how this may occur remains uncertain. It could be that the high lactate environment in the athlete results in a selective advantage for colonization by lactate metabolizing bacteria such as those of the *Veillonella* species.

Keohane et al. [36] investigated the GM response of four ultra-endurance athletes during an ocean crossing (33 days). During this event, a change in α-*diversity* was detected independently of changes in CRF, withVO_2Max_ similar pre- and post-race. Changes in taxonomic composition were observed, including an increase in butyrate-producing species (*Roseburia hominis* and members of the genus *Subdoligranulum*) and species associated with improved insulin sensitivity (*Dorea longicatena*). Bacterial species that have shown increased gene expression include *Prevotella*, as already observed by Petersen et al. [27]. Many of these changes, moreover, persisted in the following three months of follow-up. It was observed [29] that significantly increased species richness after endurance running within a single day (half marathon) in 20 healthy amateur runners in the families of *Coriobacteriaceae* e *Succinivibrionaceae* involved in the metabolism of steroids and activation of dietary polyphenols in the human gut [125]. Although α-*diversity* remains substantially stable, this finding suggested that long-distance running can promote, throughout the GM, the regulation of energy and hormone levels to maintain the homeostasis of electrolyte and gut environment.

Morishima et al. [46] investigated the relationship between intensive exercise and the GM status, in female elite endurance runners (ER), compared with the non-athletic healthy control group. They found that in the ER group, some bacteria associated with gut inflammation (*Haemophilus*, *Rothia,* and *Ruminococcus ganvus*) were more abundant. Counterintuitively, *Fecalibacterium*, known as a beneficial butyrate producer, was also more abundant in the ER group. This could be explained by the fact that an abnormal intestinal environment prompts the *Fecalibacterium* to produce succinate, a risk factor for diarrhea and loose stools [126], and not butyrate. This suggests that prolonged high-intensity exercise may lead to a form of dysbiosis in the athlete.

### 4.9. Physical Activity and Bacterial Changes in the Elderly Population

Morita et al. [37], in a non-randomized comparative study, investigated the effects of an aerobic exercise protocol on the GM composition of healthy elderly women and found a relative abundance of *Bacteroides*, associated with improvement in metabolic disease [91,127] in subjects who performed the aerobic activity, which was positively correlated with an improvement in the 6 min walking test. One study [33] attempted to assess whether endurance activity can modulate GM in elderly men and whether these changes are associated with specific cardiometabolic conditions in the host. While genus *Oscillospira*, associated with reduced BMI [128,129], increases, species *Clostridioides difficile* decreases. Furthermore, changes in these taxa were correlated with changes in several cardio-metabolic risk factors such as systolic and diastolic blood pressure. Researchers found that after six months of combined training, seven high-intensity and eight moderate-intensity exercises [47], *Oscillospira*, *Bifidobacterium,* and *Anaerostipes*, health-related genus [130,131], were increased.

Fart et al. [42] focused on identifying GM profiles related to healthy aging through GM analysis of senior orienteering athletes and found that 18% of the total fecal GM consists of *F. prausnitzii*, but no increase in microbial diversity was found, and a lower abundance of *Parasutterella excrementihominis* and *Bilophila wadsworthia* associated with decreased intestinal health [132,133,134]. An observational study [48] tried to assess the impact of endurance exercise performed four times per week, throughout life, in elderly athletes (EA), on the GM diversity and structure, compared to the control group that only met the PA recommendation for older adults [135]. Although no significant differences were found, there was a lower presence of the *Bacteroides* genus and an increased abundance of the *Prevotella* genus, as already observed in previous studies [22,40]. The small sample size is the main limitation. At least it was found that eight weeks of combined exercise protocol in untrained older women [44] led to a decrease in *Firmicutes* phylum, *Bacteroidaceae* family, and *Clostridioides Escherichia* genera, related to detrimental outcomes [136,137].

### 4.10. Physical Activity, Gut Microbiota and Overweight

Among the few longitudinal studies, Allen et al. [31] sought to investigate the effects of exercise on GM. Thirty-two sedentary, normal-weight (BMI < 25) and obese (BMI > 30) adult subjects (20–45 years) were recruited. After two weeks of baseline testing, they were administered an aerobic activity protocol (30 to 60 min, three sessions per week, moderate to intense 60–75% HRR) lasting six weeks, followed by an equal number of weeks during which the subjects did not exercise (washout). Participants completed a food diary for seven days prior to the intervention; a dietician then drew up an eating plan based on the diary, which the participants had to follow for the three days prior to the collection of stool samples, while on the remaining days they were asked to follow their usual eating patterns including the use of caffeine, alcohol, and supplements normally taken. Thus, no significant differences in β-*diversity* were found after the exercise protocol and maintained similarity even after the washout period. Aerobic activity induced a significant increase in SCFAs concentration in the feces, supporting what has previously been observed in both animals and humans [23,49], but only in normal-weight subjects; however, this effect disappeared during the washout period. The exercise seemed able to alter the abundance of certain microbial species in a BMI-dependent manner: with normal-weight subjects experiencing an increase in *Fecalibacterium* and a decrease in *Bacteroides* genus, while obese subjects experienced the exact opposite. Together with the concentration of SCFAs, many of the bacterial taxa increased with exercise and decreased following the washout period, suggesting that the effect of exercise on the GM is transient and reversible.

Munukka et al. [32] sought to determine whether an endurance protocol could affect the gut metagenome in previously sedentary overweight subjects. In order to determine individual aerobic and anaerobic thresholds, a submaximal incremental endurance test is administered, and then the 6-week (3 × week) intervention protocol is applied, with increasing intensity from light to moderate. It thus appears that this endurance protocol, in agreement with Allen et al. [31], is capable of modestly impacting the GM framework. As already observed [22,24,27], exercise increased the genus *Akkermansia*, which is inversely associated with body weight [63]. Furthermore, increased *Proteobacteria*, which may adversely affect human health [138,139], was found.

To establish whether a combined strength and endurance protocol can exert an effect on the GM of obese children (7–12 years), Quiroga et al. [40] prepared a 12-week (2 × week) concurrent training protocol through which it was found that obese subjects exhibit a bacterial profile associated with this condition. Subsequently, this protocol was shown to alter GM composition and function in obese subjects by significantly reducing the phylum *Proteobacteria*, corroborating the results obtained by Munukka et al. [32] and the class *Gammaproteobacteria*. Furthermore, an increasing trend of some bacterial genera as part of the phylum *Firmicutes* was observed, which made the GM of obese subjects like that of normal-weight control subjects. These results, therefore, suggest the presence of a negative bacterial profile related to the state of obesity that can be positively modified by PA. However, the need for further studies with a larger sample size is highlighted to increase the statistical power of these findings.

Finally, in the longest randomized controlled study to date, Kern et al. [38] set out to assess whether exercise alters the diversity, composition, and functionality of the GM in overweight or obese subjects. Specifically, the effects of regular aerobic training carried out with different modalities and intensities, but with similar energy expenditure, on the GM of 88 subjects (20–45 years old) divided into four groups: commuters traveling by non-motorized bicycles (BIKE), moderate-intensity PA (MOD) or vigorous PA (VIG) and the control group leading their usual lives (CON) are investigated over a six-month period using a specific protocol (ACTIWE) [140]. In all groups that performed PA, changes in β-*diversity* were observed, while low heterogeneity was found in the VIG group. In the MOD group, changes in α-*diversity* were noted. These discrepancies between studies may be due to the lack of a non-exercising control group and the small sample size used in previous studies. Similarly, since the primary objective of the researchers in this study is to evaluate the impact of prescribing exercise to overweight or obese subjects in a daily life context, the dietary intake was not voluntarily standardized, which makes it difficult to determine whether the observed results are due to the exercise per se or its effect on the subjects’ diet.

### 4.11. The Gut–Brain Axis Functional Basis and the Impact of Physical Activity

Only recently a connection between GM and host brain function has been identified. This relationship seems to be the result of the ability of different bacterial species to exert an influence on brain activity [141] through neuroendocrine pathways probably developed initially to connect the gut and the areas of the brain responsible for stimuli and the control of hunger/satiety and, subsequently, extended to the influence on emotions linked to different eating behaviors [142] mediated by a particular neurotrophic factor such as Brain-Derived Neurotrophic Factor (BDNF), whose concentration can be modulated by the GM [143]. This two-way communication pathway between the enteric (ENS) and central nervous systems (CNS) is known as the gut–brain axis (GBA). Several papers in recent years have highlighted the importance of this relationship as reported by Erny et al. [144], who note that host bacteria regulate the maturation and function of microglia, immune defense cells in the CNS and that impairment of these cells can be rebalanced to some extent by a diverse and complex microbiota. Furthermore, it seems that the microbiota is necessary for the normal development of the amygdala both structurally and functionally and may contribute to several brain functions dependent on this nucleus, from pain sensitivity to social behavior via emotion regulation [145].

Recent evidence suggested that PA, particularly of an aerobic nature, improving the diversity and abundance of certain bacterial genera may have positive effects on the gut and neurology [20]. The bi-directional communication between the gut and the brain has embryogenic origins as the CNS and the ENS, or the metasympathetic system, have identical tissue derivation [146]. Communication between these two systems is achieved through neuroendocrine, immune, and neuronal regulation [147] and, in particular, is mediated through signals traveling via the vagus nerve and the hypothalamic-pituitary-adrenal (HPA) axis [148]. Through these pathways, alterations in the gut microbiota can influence the communication that takes place between the gut and the brain [20]. Indeed, several molecules produced by the GM, such as SCFAs, bile acids, and tryptophan, interact with enteroendocrine cells by propagating a long-distance signal and activating the vagus nerve [149]. For example, if the information received centrally from the vagus nerve indicates an imbalance in the composition of the GM, the CNS decides what action to take, usually adverse to the health of the host. Conversely, supplementation with specific probiotics has been shown to improve brain function by promoting an anxiolytic effect, which is lessened by vagotomy [143]. Similarly, GM via LPS can damage the intestinal mucosa, which promotes the release of pro-inflammatory cytokines capable of hyper-stimulating the HPA [150], a condition associated with psychological disorders [151]. Several behaviors appear to be influenced by GM, which depend on a properly functioning serotonergic neurotransmission; it should be noted that 90% of serotonin, known as the main cognitive and mood regulator [152], can be synthesized and regulated by specific gut bacteria [149].

### 4.12. Physical Activity, Gut Microbiota and Cognitive Ability

According to Gubert et al. [153], considering the concomitant intestinal dysbiosis present in several neurodegenerative diseases and the impact that PA or exercise can have on the intestinal microbiome and neuronal degeneration, a triangulation between these aspects seems plausible to assess whether PA can contribute to the modulation of neurodegeneration through GM. Among the few studies that tried to find a correlation between PA and GM in humans, none have investigated this neural aspect in-depth, whereas some studies in animal models have shown some initial evidence.

Kang et al. [52] showed that mice subjected to a 16-week training protocol reported an improvement in memory associated with the increase in the *Firmicutes:Bacteroidetes* ratio induced by exercise. They also point out that one hour per day of exercise can increase the relative abundance of the family *Lachnospiraceae*, which is negatively correlated with anxiogenic behavior and is capable of producing butyrate (SCFA), a molecule that over-regulates BDNF expression in the hippocampus and frontal cortex, supporting the survival of existing neurons and stimulating the formation of new neurons and synapses [154]. Thus, according to the authors, this association between induced changes in certain GM phyla and families and memory improvement could be used as a biomarker for exercise-induced effects at a cognitive level.

Feng et al. [62] considered a condition known as postoperative cognitive dysfunction (POCD), whereby the memory and learning of individuals undergoing surgery decline, particularly in the elderly [155]. Applying a treadmill training protocol during the six weeks prior to surgery, performed five days a week, prevented POCD, assessed through the Morris Maze Test, along with an increase in microbial diversity (α and β) and improvement in the *Firmicutes:Bacteroidetes* ratio in favor of the former, indicating an improvement in dysbiosis following surgery. Concomitantly, the ability of exercise to reduce the postoperative neuroinflammatory state is observed. Thus, the authors suggested that GM contributed to the modulation of inflammatory status through a pathway mediated by the exercise performed previously since a low GM diversity appears associated with increased inflammation [156]. According to Gubert et al. [153], experimental data have shown that GM modifications induced by an aerobic activity, specifically characterized by an increase in certain bacteria (e.g., *Lactobacillus plantarum* and *Streptococcus thermophile*), are capable of inducing the synthesis of serotonin, a molecule that protects against symptoms of anxiety and depression [157].

Abraham et al. [65] extrapolated findings that seem to indicate the ability of exercise to improve cognitive function and some markers of Alzheimer’s disease (AD), neurodegenerative disease for which there is currently no cure. They hypothesize that regular exercise combined with nutritional intervention may reduce the incidence of AD. Using transgenic mice (APP/PS1), which mimic a model of AD, subjected to a 20-week treadmill exercise protocol, they note a significant improvement in the Morris Maze Test, which assesses spatial memory, and a reduction in β-amyloid plaques, one of the main aspects involved in AD [158], supporting what was previously observed by Lin et al. [159]. Near these plaques, an increase in microglia, important for brain development by providing structural and metabolic support to neurons and involved in neuroplasticity and regulation of neural repair [160], is found, underlining the neuroprotective effect of exercise. This appears to be associated with the abundance of some bacterial strains (*Eubacteria*, *Roseburia*) and the reduction of others, so these results suggest that the cognitive effects of exercise may be mediated through GM alteration, reducing the levels of microbes involved in disease exacerbation and promoting the abundance of those bacteria capable of producing SCFAs that appear beneficial. Similar bacterial changes were highlighted in a meta-analysis that sought to identify a relationship between GM and Parkinson’s disease (PD). The ecological imbalance of the GM can induce IP, impaired SCFA production, and altered immunity function. Therefore, alteration of the GM could be considered an environmental trigger of the pathological process of PD and contribute to its development [161]. This would seem to support the so-called dual-hit hypothesis that PD is due to peripheral dysregulations of the GM, known as dysbiosis. This makes it possible to speculate that the benefits of exercise observed in PD may be partly due to its ability to restore GM [162].

## 5. Limitations and Future Directions

Despite the promising future and interest in this research area, multiple aspects make the development of adequate study protocols complicated, first and foremost the not easy dissociation between PA and nutrition. Furthermore, in preclinical studies, the same type of diet (LFD, HFD, LC, HC) provided to animals may interfere differently with the GM depending on the type of diet used, from the more classic standard chow, which has manufacturer-dependent variability, to the purified diet. The studies carried out on humans appear to be more difficult to interpret, first of the difficulty of dietary control also due to the instruments used for this purpose that are imprecise, starting from the most accurate modality, the food diary, that can psychologically lead subjects to eat better, to the most used instruments, the FFQ, subject to a high degree of uncertainty. Different types of bacterial genome sequencing methods and different types of PA used in the studies make it difficult to extrapolate guidelines. Future research should isolate more confounding elements, first and foremost the dietary aspect, with a greater focus on young and elderly populations, as well as on resistance training protocols where evidence is scarce. Specific attention should be paid to the development of research in humans to investigate the concrete impact of PA on the GBA.

## 6. Conclusions

Although the literature analyzing GM is steadily increasing, the effects of PA on bacterial flora remain uncertain. A PA or exercise performed voluntarily appears to attenuate intestinal inflammation, in contrast to a forced activity that instead increases this condition in animal models. Vigorous endurance exercise can negatively affect the GM framework in humans; furthermore, only aerobic activities can alter the GM structure, probably because of the positive correlation between CRF and microbial diversity.

PA can stimulate bacterial community richness by altering SCFAs-producing species, as well as favoring the colonization of health and athletic performance-promoting strains (e.g., *A. muciniphila* and *Veillonella*). The ability to promote a bacterial composition capable of protecting the intestinal mucosa from possible permeability and counteracting HFD-induced changes in the GM is observed.

Overall, PA can increase the relative abundance of phylum *Bacteroidetes* and reduce that of *Firmicutes* as well as may exert greater and more lasting benefits if undertaken from an early age, but the effects on GM seem to gradually disappear when the PA is no longer practiced.

PA seems able to regulate cognitive conditions (e.g., anxiety and depression) and functionality (e.g., Alzheimer’s and Parkinson’s Disease) through modifications of microbial composition and subsequently the production of certain protective molecules, to date in animal models.

## Figures and Tables

**Figure 1 nutrients-14-03293-f001:**
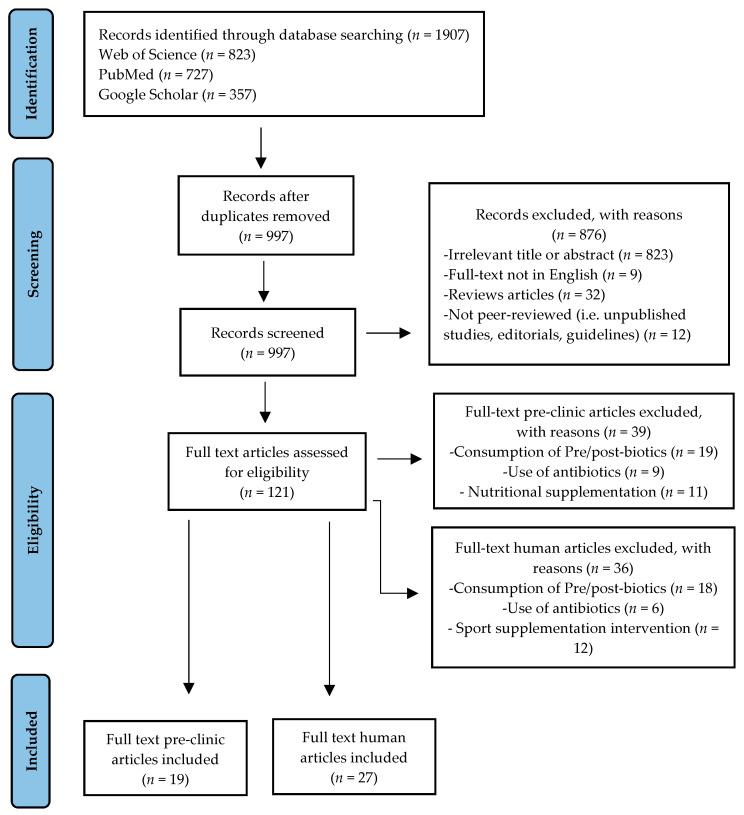
Study selection and eligibility screening flow diagram.

**Table 1 nutrients-14-03293-t001:** Summary characteristics of reviewed human studies.

Authors	Study Design	Sample	Subjects Age (Years)	Type PA	Protocol	Diet Assessment	Duration Intervention	Main Outcomes
Clarke et al., 2014 [22]	Cross-sectional	*n* = 86 (M) elite professional rugby players (*n* = 40) (BMI 29.1 ± 2.9), healthy control (*n* = 46) (23: BMI ≤ 25—23: BMI > 28)	Elite: 29 (±4) Control: 29 (±6)	Rugby	/	187-food items FFQ. Macronutrients, fiber, and supplement intake	/	Athletes: ↑ α-*diversity*, ↑ diversity Firmicutes (phylum), ↑ Prevotella, ↓ Bacteroides, ↓ Lactobacillus Athletes/Low BMI: ↑ Akkermansia (genus)
Estaki et al., 2016 [23]	Cross-sectional	*n* = 39 (M/F) healthy subjects, stratified by CRF (Low; Average; High)	L: 25.5 (±3.3) A: 24.3 (±3.7) H: 26.2 (±5.5)	Aerobic (Mixed activities)	/	24 h dietary recall interview. Macronutrients, fiber, saturated fat, and PUFA intake	/	VO_2_ peak positively associated with ↑ GM diversity; ↑ CRF = ↑ taxa producers SCFAs. No differences in α and β-*diversity*
Bressa et al., 2017 [24]	Cross-sectional	*n* = 40 (F) active (ACT) (*n* = 19) and sedentary (SED) (*n* = 21) subjects, defined by WHO recommendations	ACT: 30.7 (±5.9) SED: 32.2 (±8.7)	Aerobic (Mixed activities)	/	97-food items FFQ. Macronutrients, fiber, and main food intake	/	ACT: PA ↑ health-promoting bacteria (F.prausnitzii, R.hominis, A.muciniphila) SED: ↑ Barnesiellaceae, ↑ Turicibacter, ↓ Cropococcus No differences in α/β-*diversity* and at phylum level between groups.
Mörkl et al., 2017 [25]	Cross-sectional	*n* = 106 (F) Anorexia nervosa (AN) patients (*n* = 18), normal weight (NW) (*n* = 26), overweight (OW) (*n* = 22), obese (O) (*n* = 20) and athletes (AT) (*n* = 20)	24.5 (±4.6)	Ball sports	/	Two 24 h recalls. Macronutrients, fiber, Vit D, and magnesium intake	/	↓ GM α-*diversity* in obese and AN groups compared to athletes.
Yang et al., 2017 [26]	Cross-sectional	*n* = 71 (F) premenopausal with low (L), moderate (M), high (H) CRF	L: 40.4 (36.9–44.0) M: 39.7 (35.5–43.8) H: 30.6 (25.6–35.6)	Aerobic (Mixed activities)	/	3-days food records (2 weekdays, 1 weekend day). Macronutrients and total energy intake.	/	↓ Bacteroides and ↑ Eubacterium rectale–clostridium coccoides in Low VO_2max_ compared to High VO_2max_ group.
Petersen et al., 2017 [27]	Cross-sectional	*n* = 33 (M/F) professional (*n* = 22) and amateur (*n* = 11) level competitive cyclists	19–49 (Median age 33)	Cycling	/	Food questionnaire. Macronutrients and alcohol intake.	/	No significant correlations between taxonomic cluster and professional or amateur level. ↑ Prevotella relative abundance in cyclists training >11 h/wk
Paulsen et al., 2017 [28]	Pilot study	*n* = 12 (F) BCS subjects engaging less than 30′ of vigorous or 60′ of moderate-intensity PA per wk in previous 6 months	55 (±13)	Aerobic (Mixed activities)	Gradually increase participants to ≥150 weekly minutes of moderate intensity.	3-day diet record	3 months	Significant differences in β-*diversity* were found for CRF suggesting changes in specific taxa present (↑ Roseburia ↑ SMB53 subset of family Clostridiaceae)
Zhao et al., 2018 [29]	Cross-sectional	*n* = 20 (M/F) health amateur runner	31.3 (±6.1)	Endurance	Half-marathon	Dietary questionnaire. Macronutrient intake	/	After running no changes in α-*diversity*. ↑ Coriobacteriaceae and Succinivibrionaceae families. ↓ Ezakiella and Romboutsia genus, ↑ Coprococcus, Actinobacillus and Ruminococcus genus
Barton et al., 2018 [30]	Cross-sectional	*n* = 86 (M) elite professional athletes (*n* = 40), healthy control (*n* = 46) (22: BMI ≤ 25.2–24: BMI ≥ 26.5)	Elite: 29 (±4) Control: 29 (±6)	Rugby	/	187-food items FFQ. Macronutrients and total energy intake.	/	↑ Pathways (↑ AA biosynthesis, ↑ carbohydrate metabolism) and ↑ fecal metabolites (microbial produced SCFAs) in athletes
Allen et al., 2018 [31]	Longitudinal	*n* = 32 (M/F) previously sedentary subjects, lean (*n* = 18) and obese (*n* = 14)	Lean: 25.1 (±6.52) Obese: 31.14 (±8.57)	Aerobic (cycling or running)	30′ to 60′ 3 × wk moderate-to-vigorous intensity (60–75% HRR) exercises	7-days dietary records, 3-days food menu before each fecal collection. Macronutrient, micronutrient, and total energy intake	6 weeks	No β-*diversity* differences among groups. ↑ SCFAs producing taxa related to BMI (Fecalibacterium: ↑ lean ↓ obese, Bacteroides: ↓ lean ↑ obese). Changes largely reversed after 6 wk of inactivity.
Munukka et al., 2018 [32]	Non-randomized trial	*n* = 17 (F) sedentary subjects BMI > 27.5 kg/m^2^	36.8 (±3.9)	Endurance (cycling)	40′ to 60′ 3 × wk exercises, low to moderate intensity	3-days food records (2 weekdays and 1 weekend day). Macronutrients, fiber, and total energy intake	6 weeks	↑ Akkermansia and ↓ Proteobacteria (exercise-responsive taxa). Changes in GM do not affect systemic metabolites. No differences in α-*diversity*, slight ↑ β-*diversity*
Taniguchi et al., 2018 [33]	Randomized crossover trial	*n* = 33 (M) elderly Japanese subjects	62–76	Endurance (cycling)	3 × wk ce, 30′ (wk 1/2)—45′ (wk 3/5), with incremental intensity	Self-administered FFQ, semi-weighted 16-days dietary records. Macronutrients and total energy intake.	5 weeks	No differences in α and β-*diversity*. ↓ C.difficile, ↑ Oscillospira. Minor changes in GM associated with cardiometabolic risk factors.
Durk et al., 2019 [34]	Cross-sectional	*n* = 37 (M/F) healthy subjects	25.7 (±2.2)	Aerobic (running)	/	Instructed to follow their normal diet for 7-days and MyFitnessPal app tracking. Macronutrients, fiber, coffee, alcohol, and total energy intake.	/	VO_2max_ positively associated to ↑ Firmicutes:Bacteroidetes ratio. No differences in α and β-*diversity*.
Scheiman et al., 2019 [35]	Cross-sectional	*n* = 25 (M) subjects. Athletes from the Boston Marathon (*n* = 15), sedentary controls (*n* = 10)	/	Endurance	Marathon	questionnaire and daily annotation sheet	/	↑ Veillonella relative abundance, in marathon runners post marathon, which can positively influence running performance through the conversion of lactate.
Keohane et al., 2019 [36]	Observational	*n* = 4 (M) ultra-endurance athletes	26.5 (±1.3)	Endurance	Trans-oceanic rowing	FFQ and MyFitnsessPal mobile application	33-day event and 3 months follow-up	↑ α-*diversity* throughout event. ↑ abundance of butyrate producing species (i.e., Roseburia) and species associated with improved metabolic health (Dorea longicatena). Many of the adaptions in GM structure and metaproteomics persisted at 3 months follow-up.
Morita et al., 2019 [37]	Non-randomized comparative trial	*n* = 32 (F) healthy sedentary elderly subjects, trunk muscle (TM) (*n* = 14) and aerobic exercise (AE) (*n* = 18) intervention	70 (66–75)	Aerobic or anaerobic	TM: 1 h weekly resistance training AE: 1 h daily brisk walking ≥ 3 METs	138-food and beverage items FFQ. Macronutrients, fiber, saturated fat and total energy intake.	12 weeks	↑ Bacteroides relative abundance only in the AE group.
Kern et al., 2020 [38]	Randomized controlled trial	*n* = 88 (M/F) overweight/obese subjects, moderate intensity (*n* = 31) (MOD), vigorous intensity (*n* = 24) (VIG), bicycling (*n* = 18) (BIKE), control (*n* = 14) (CON)	36 (30; 41) Median (25th percentile; 75th percentile)	Aerobic (MOD&VIG: walking/running, cycling, stepping. BIKE: cycling)	MOD: 5 × wk LTPA at 50% VO_2peak_ VIG: 5 × wk LTPA at 70% VO_2peak_ BIKE: 5 × wk active bicycle commuting to and from work (F: 9–15 km/M: 11–17 km daily), self-selected intensity	Food registrations (3 weekdays—1 weekend day), participants were asked to weigh and register intake of food and beverages. Macronutrients, fiber, and total energy itnake.	6 months	β-*diversity* changed in all groups compared to CON, ↑ α-*diversity* in VIG compared to CON. Decreased heterogeneity in VIG. No genera changed significantly.
Catellanos et al., 2020 [39]	Cross-sectional	*n* = 109 (M/F) healthy subjects, active (*n* = 64) (ACT) and sedentary (*n* = 45) (SED), described by WHO recommendations	ACT: 32.17 (±7.40) SED: 33.69 (±7.96)	Aerobic (Mixed activities)	/	93-food items FFQ. Macronutrients, fiber, ethanol, and total energy intake.	/	GM network of active people has higher efficiency and transmissibility rate. Key bacteria reorganization from ACT to SED: *Roseburia fecis*, unclassified roseburia spp. Key bacteria reorganization from SED to ACT: unclassified *Sutterella* spp.
Quiroga et al., 2020 [40]	Randomized controlled trial	*n* = 39 obese pediatric children (*n* = 25) and healthy control (*n* = 14)	7–12	Endurance plus strength	2 × wk combined endurance (sprint of 30″ max cadence at 3′30″, 4′30″, 5′30″, and 6′30″) and strength training (30–70% 1 RM)	Nutritional advice for a healthy and balanced diet.	12 weeks	↓ Proteobacteria phylum and Gammaproteobacteria class, ↑ Blautia, Dialister and Roseburia genera lead to a GM profile like that of healthy children.
Rettedal et al., 2020 [41]	Non-randomized trial	*n* = 29 (M) overweight (*n* = 15) and lean (*n* = 14) subjects	Overweight: 31 (±2) Lean: 29 (±2)	Aerobic (cycling)	3 × wk ce HIIT, 60″ cycling intervals at VO_2peak_ workload interspersed with 75″ rest, 8 to 12 intervals	FFQ for baseline intake. Instructed to maintain normal dietary pattern. Macronutrients, fiber, saturated fat, PUFA, and total energy intake.	3 weeks	No differences in α and β-*diversity*. Significant association between the abundance of bacterial spp. (Coprococcus_3, Blautia, Lachnospiraceae_ge, Dorea) and insulin sensitivity marker in the overweight group.
Fart et al., 2020 [42]	Cross-sectional	*n =* 98 (M/F) older adults. community-dwelling older adults (CDO) (*n* = 70) and senior orienteers (SO) (*n* = 28)	CDO: 72 SO: 68.5	Orienteering	/	FFQ	/	In SO group compared to CDO group: ↑ F.prausnitzii. No enhanced microbial diversity. ↓ Parasutterella excrementihominis and Bilophila wadsworthia, associated with decreased intestinal health.
Bycura et al., 2021 [43]	Non-randomized trial	*n* = 56 (M/F) healthy students, cardiorespiratory exercise (*n* = 28) (CRE), resistance exercise (*n* = 28) (RTE)	CRE: 20.54 (1.93) RTE: 21.28 (3.85)	Aerobic or anaerobic	CRE: 1 h, 3 × wk (2-days group cycling, 1-day rotating CRE activity) 60–90% HR_max_ RTE: 1 h 3 × wk full/lower/upper body at 70–85% 1 RM	Not controlled or recorded. Instructed to maintain their typical dietary practice and report major deviations.	8 weeks	CRE: initial changes to GM (wk 2, 3) not sustained through or after the intervention. RTE: no changes in microbiome composition.
Zhong et al., 2021 [44]	Randomized controlled trial	*n* = 12 (F) previously inactive older healthy subjects, exercise (*n* = 6) and control (*n* = 6)	Exercise: 69.83 (±4.50) Control: 67.50 (±4.28)	Aerobic (stepping) and anaerobic	1 h 4 × wk combined aerobic and resistance exercises (progressive overload)	Not controlled or recorded	8 weeks	No changes in α-*diversity*. ↑ Prevotella, ↑ Verrucomicrobia, ↓ Proteobacteria abundance in the exercise group.
Moitinho-Silva et al., 2021 [45]	Randomized controlled trial	*n* = 36 (M/F) healthy physical inactive subjects, endurance (*n* = 12) and strength exercises (*n* = 13) with control (*n* = 11). Elite athletes for comparison (*n* = 13)	Endurance: 31.4 (±8.3) Strength: 29.9 (±7.9) Control: 33.4 (±7.9) Elite: 30 (±9.9)	Aerobic or anaerobic	Endurance: 30′ (at least) 3 × wk running Strength: 30′ 3 × wk whole-body hypertrophy strength training	Food questionnaire Elite: no data. Macronutrients, fiber, and total energy intake.	6 weeks	No specific bacteria changes. GM change patterns largely varied among individuals of the same group. No differences in α-*diversity* between elite and physical inactive subjects.
Morishima et al., 2021 [46]	Cross-sectional	*n* = 29 (F) subjects. Endurance runner (*n* = 15) and healthy non-athletic (*n* = 14)	Runners(R): 20.5 (±1.2) Control(C): 20.9 (±0.3)	Endurance (running)	/	/	/	In ER group: ↑ Haemophilus, Rothia and Ruminococcus gnavus genus, associated with gut inflammation.
Erlandson et al., 2021 [47]	Pilot study	*n* = 15 (M/F) sedentary older adults	58 (±8.0)	Aerobic (walking) and anaerobic	20/30′ aerobic exercise + 3 sets × 8 reps of resistance exercise at low intensity: 3 × wk ~50′ session	3-day diet record. Macronutrient intake	24 weeks	↑ Bifidobacterium, Oscillospira and Anaerostipes, associatet to gut health benefits. ↓ Prevotella and Succinivibrio, associated to inflammatory states.
Šoltys et al., 2021 [48]	Cross-sectional	*n =* 22 (M) elderly subjects. Lifetime endurance athletes (*n* = 13) and healthy control who met ACSM PA recommendation (*n* = 9)	LA: 63.5 CTRL: 64.9	Endurance (cycling)	/	24 h dietary recording over five consecutive days	/	In LA group comparet to CTRL group: no differences in α-*diversity*. ↓ Bacteroidetes genus ↑ Prevotella genus.

M: male; F: female; BMI: body mass index; ↑: increase; ↓: decrease; GA: gene amplification; FFQ: food frequency questionnaire; CRF: cardiorespiratory fitness; VO_2peak_: peak oxygen uptake; GM: gut microbiome; SCFAs: short-chain fatty acids; wk: week/s; PA: physical activity; IPAQ: international physical activity questionnaire; WHO: world health organization; CI: confidence interval; VO_2max_: maximal oxygen uptake; HHR: heart rate reserve; ce: cycle ergometer; LTPA: leisure-time physical activity; METs: metabolic equivalent of task; HIIT: high-intensity interval training; T2D: type 2 diabetes; BCS: Breast cancer survivors; HR_max_: maximal heart rate; rpm: revolutions per minute; PUFA: polyunsaturated fatty acids.

**Table 2 nutrients-14-03293-t002:** Summary characteristics of reviewed pre-clinic studies.

Authors	Study Design	Sample	Species	Type PA	Protocol	Duration Intervention	Main Outcomes
Matsumoto et al., 2008 [49]	Randomized block design	*n* = 14 (M) exercise (*n* = 7) and sedentary control (*n* = 7)	Wistar rats (6 wk old)	Aerobic	VWR	5 wk	VWR group: ↑ Butyrate (SCFA); ↑ Butyrate-producing bacteria, phylum Firmicutes (SM/11, T2-87)
Choi et al., 2013 [50]	Randomized controlled trial	*n* = 12 (M) exercise (*n* = 6) and sedentary control (*n* = 6)	C57BL/6 mice (11–13 months old)	Aerobic	VWR	5 wk	VWR group: ↑ phylum Firmicutes (i.e., lactobacillales order), ↓ phyla Tenericutes and Bacteroidetes. Changes in microbiota induced by PCBs exposure were attenuated.
Queipo-Ortuño et al., 2013 [51]	Case-control study	*n* = 40 (M) ABA exercise (*n* = 10) ABA sed (*n* = 10) AL exercise (*n* = 10) AL sed (*n* = 10)	Sprague-Dawley rats (5 wk old)	Aerobic	VWR	6 days	AL exercise group: ↑ Lactobacillus, Bifidobacterium and Blautia. ↑ Organic acid lactate converted in butyrate (SCFA) ↓ Clostridium and Enterococcus.
Kang et al., 2014 [52]	Randomized controlled trial	*n* = 40 (M) ND (*n* = 10) ND exercise (*n* = 10) HFD (*n* = 10) and HFD exercise (*n* = 10)	Wild-type mice (8 wk old)	Aerobic	FWR 1 h at 7 m/min × 5 days/wk	16 wk	Exercise alone caused great changes in gut microbiota: ↑ Firmicutes, Proteobacteria and Actinobacteria phyla. ↓ Bateroidetes phylum.
Evans et al., 2014 [53]	Randomized controlled trial	*n* = 48 (M) LF sed (*n* = 12) LF exercise (*n* = 12) HF sed (*n* = 12) HF exercise (*n* = 12)	Wild-type mice (5 wk old)	Aerobic	VWR	12 wk	Exercise induced unique change in gut microbiota: ↑ Bacteroidetes and ↓ Firmicutes phylum; ↓ Actinobacteria preventing DIO
Petriz et al., 2014 [54]	Prospective cohort study	*n* = 15 Obese (*n* = 5), hypertensive (*n* = 5) and high blood pressure (*n* = 5)	Wistar rats	Aerobic	Treadmill 30′/day × 5/days/wk. speed progressively increased	5 wk	In hypertensive: ↑ Frimicutes ↓ Proteobacteria ↑ Lactobacillus ↑ Allobaculum. In obese: ↑ Pseudomonas and lactobacillus. In all groups: ↑ Firmicutes and ↓ Proteobacteria
Allen et al., 2015 [55]	Randomized controlled trial	*n* = 29 (M) VWR (*n* = 10) FTR (*n* = 10) sed control (*n* = 9)	C57BL/6J mice	Aerobic	VWR vs. FTR (40′ × 5 days/wk)	6 wk	In VWR group: ↓ Turicibacter In both groups: ↔ Bacteroidetes and Firmicutes, ↓ bacterial richness.
Hsu et al., 2015 [56]	Prospective cohort study	*n* = 24 (M) SPF (*n* = 8) GF (*n* = 8) and BF (*n* = 8) gnotobiotic mice	C57BL/6JNarl mice (12 wk old)	Endurance	Swimming	/	In GF and BF groups: ↓ SCFAs ↓ Gpx and ↓ CAT. Gut microbial status can be crucial for physical performance linked to antioxidant enzyme systems.
Liu et al., 2015 [57]	Prospective cohort study	*n* = 30 (F) HCR-EX (*n* = 8) HCR-SED (*n* = 7) LCR-EX (*n* = 8) LCR-SED (*n* = 7) OVX rats	(26 wk old)	Aerobic	VWR	11 wk	In HCR-ex group: ↑ Firmicutes In LCR-ex group: ↓ Firmicutes, ↔ Bacteroidetes.
Mika et al., 2015 [58]	Randomized controlled trial	*n* = 40 (M) juvenile (*n* = 20) and adults (*n* = 20)	F344 rats	Aerobic	VWR	6 wk	In juvenile rats compared to adults: ↓ Firmicutes ↑ Bacteroidetes ↑ Bacterial genera.
Campbell et al., 2016 [59]	Case-control study	N = 36 (M) LS (*n* = 9), DIOS (*n* = 9) Lex (*n* = 9) and DIOex (*n* = 9) mice	C57BL/6NTac mice (6 wk old)	Aerobic	VWR	12 wk	In DIOex and Lex groups: ↑ Fecalibacterium prausnitzii
Denou et al., 2016 [60]	Randomized controlled trial	*n* = 16 (M) exercise (n = 8) and untrained (*n* = 8) mice	C57 BL/6 mice (8 wk old)	Aerobic (HIIT)	1 h of treadmill running 3 days/wk	6 wk	In exercise group: ↓ Firmicutes:Bacteroidetes ratio.
Lamoureux et al., 2017 [61]	Prospective cohort study	*n* = 42 (M/F) voluntary exercise (*n* = 10), forced exercise (*n* = 11) and non-exercise control (*n* = 21)	C57BL/6 mice (6/10 wk old)	Aerobic	VWR vs. FTR	8 wk	In all groups: ↑ Rikenellaceae and Lachnospiraceae; ↔ species richness.
Feng et al., 2017 [62]	Randomized controlled trial	*n* = 14 (M) Surgery (*n* = 6) and Sham (*n* = 8) group	HCR and LCR rats	Aerobic	Treadmill	6 wk	In LCR: ↑ α-*diversity* In LCR and HCR: ↑ β-*diversity* In HCR: ↑ Firmicutes and ↓ Bacteroidetes. Exercise prevented POCD
Yuan et al., 2018 [63]	Randomized controlled trial	*n* = 20 (M) excessive swimming (ES) (*n* = 10) and non-swimming (NS) (*n* = 10)	Kunming (KM) mice (6 wk old)	Endurance	Swimming until exhaustion	4 wk	In ES group: ↓ microbial diversity; ↓ Bacteroidales (S24-7) and Lachnospiraceae; ↑ Helicobacteraceae family. ↑ Helicobater and Bacteroides, ↓ Odoribater genus.
Ribeiro et al., 2019 [64]	Randomized controlled trial	*n* = 40 (M) SDC (*n* = 10) SDT (*n* = 10) HFC (*n* = 10) and HFT (*n* = 10)	C57BL6 mice	Aerobic	30′ treadmill running 5 days/wk	8 wk	In HFT group: ↑ Proteus genus In HFT and SDT: ↑ Vagococcus genera. No significant changes in gut microbiota structure.
Abraham et al., 2019 [65]	Prospective cohort study	*n* = 32 (M) exercise (*n* = 6) Fram (*n* = 6) and Combined (*n* = 6) and control (*n* = 14)	APP/PS1 transgenic mice	Aerobic	Treadmill 4 days/wk × 60′	20 wk	In exercise group: ↑ butyrate bacterial producing (Marvinbryantia formatexigens and Butyrivibrio pro teoclasticus) ↑ Clostridium Eubacterium and Roseburia; ↓ L. johnsonii.
Leigh et al., 2020 [66]	Case-control study	*n* = 48 (M) Csed (*n* = 12) Cex (*n* = 12) Cafsed (*n* = 12) Cafex (*n* = 12)	Sprague-Dawley rats (6/7 wk old)	Aerobic	FTR	4 wk	In all groups: No significant change in the overall composition of gut microbiome
Li et al., 2021 [67]	Case-control study	*n* = 54 (M) SDsed (*n* = 6) SDex (*n* = 6) and HFsed (*n* = 6) HFex (*n* = 6)	C57BL/6J mice (12 wk old)	Aerobic	VWR	4 wk	In SD groups: ↔ richness In HFex group: ↑ richness; ↓ Firmicutes:Bacteroidetes ratio; ↑ Bacteroidales S24-7; ↑ Prevotellaceae ↑ Bifidobacteriaceae.

VWR: voluntary wheel running; ↑: increase; ↓: decrease; ↔: no changes; FWR: Forced wheel running; ABA: activity-based anorexia; AL: ad libitum; ND: normal diet; HFD: high fat diet; LF: low fat; HF: high fat; DIO: diet-induced obesity; CHILD: childhood; FTR: forced treadmill running; sed: sedentary; SPF: Specific pathogen-free; GF: germ free; BF: Bacteroides fragilis; HCR: high capacity running; LCR: low capacity running; OVX: ovariectomy; C: chow diet; Caf: cafeteria diet; Fram: Frameline; D: standard diet; POCD: postoperative cognitive dysfunction.

## Data Availability

No new data were created or analyzed in this study. Data sharing is not applicable to this article.

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
