# Peer review of "The Effects of Physical Activity on the Gut Microbiota and the Gut–Brain Axis in Preclinical and Human Models: A Narrative Review"

_nutrients, 2022, doi:10.3390/nu14163293_

Round 1
Reviewer 1 Report
Dear Authors,
I received a manuscript for review for the title: „The Impact of Physical Activity and Exercise on the Gut Microbiota, Physical Performance and Cognitive Functions in Preclinic and Human Models: A Narrative Review“
It is, as the title suggests, a Narrative Review, however, to my mind the work is not clear and insightful enough and the reader gets lost in the multitude of results. I suggest extensive modifications to the thesis:
I firmly recommend the following to be added to the methodology:
- the number of studies found, the number of studies excluded (including justification) and the final number of studies included in the review - I recommend presenting this in graphical form
- an indication that the studies involved both human (specify the number) and animal (detto)
- provide a clearer and more explicit rationale for the selection of the included publications. This does not need to be stated in the methodology, but can be included in each subchapter of the results section
I further recommend addition of the following to the results:
- analysis of approaches and results obtained
- evaluation of the selected articles and their synthesis
I suggest using comprehensively organized tables in this section, not merely a text in which the reader is getting lost.
The Narrative Review is also supposed to have a discussion part, which is omitted in the text, I suggest the addition of it. In it, the authors are expected to identify the contradictions in the results of the researches reviewed, they should come to more general conclusions and recommendations.
Finally, please unify the listing of authors in the References section
Reviewer 2 Report
Dear authors,
Thank you very much for your paper, I believe it is very interesting. I really enjoy your explanation of gut microbiota. However, I believe that the preclinic studies could be used to support the human studies and not as a separate point.
Here you have some comments to help you to improve your paper from my area of expertise (Exercise)
Comment about terminology:
Exercise is included inside PA, please read (Corbin, Pangrazi, & Franks, 2000), I do understand the confusion because I have read a paper that says physical exercise in a paper about GM, but in the field of exercise, physical education, we refer to PE for physical education and not for physical exercise. There is no need to say physical exercise, you can just say exercise.
Comments about grammar:
In general, when we talk about previous research it should be in past sentence. See here some examples. Line: 452 “look” should be in past simple also the same in line 474 for investigate, Also, in lines 489,514,516…. Please check the whole document.
Comments about edition:
Line 108: Joshua Lederber needs reference.
Line 384: Rugby should be added to the kind of players for a better understanding.
Line 428: CRF should be explained the first time in mention in line 428 and not in lines 432-433.
Line 446: I believe the authors mean high aerobic fitness but I’m not sure.
Line 446: What do the authors mean by aerobic training (CRE)? What does CRE stand for?
Line 447: Same for TRE
Comments about the content:
Line 502. CRF is not something that can be changed during a race, maybe the authors mean VO2? I´m not sure.
Line 445, it seems that the 130, and 131 papers are not related with exercise, so I do not understand why they are cited in this line.
Lines 608-610. I do really like this sentence “These results, therefore, suggest the presence of a negative bacterial profile related to the state of obesity that can be positively modified by PE.” although I’m not sure about the grammar and I would say PA instead of PE. For readers like me that are not expert on the field on Gut Microbiota it may be interesting to have a statement like this after each part to summarize what has been presented.
The conclusions should be easier to follow. The sentences are too long. Maybe they could be divided in different paragraph/conclusions.
In summary, I believe that the content and therefore the title of this article should be change to something shorter and easier to understand based on my comments.
Corbin, C. B., Pangrazi, R. P., & Franks, B. D. (2000). Definitions: Health, fitness, and physical activity. President's Council on Physical Fitness and Sports Research Digest.
Round 2
Reviewer 1 Report
Dear Authors,
Thank you for acknowledging my comments. In its current form, the article is, from my point of view, more readable and insightful.
I have no further comments on the final version of the article.
Author Response
Dear reviewer,
thank you very much for your valuable comments
Reviewer 2 Report
Dear authors
I have really noticed the improvement of your paper. The table has helped me to understand it better, however in the column of type of PA, you are mixing type with intensity, type will be rugby, running, cycling… and intensity will be aerobic, anaerobic… For example, you can cycle at aerobic or anaerobic intensity. It will preferable if you put both type and intensity in the table. For example, for rugby, there is not need to put the intensity, but when you say aerobic, it will be interesting to know the kind of activity, running, swimming, walking…
The conclusion has very much improved, however, you should be careful with the acronyms, not only in the conclusion but in the whole paper. In line 621 you use PA and in line 622 you use physical activity.
Kind regards,
